# Cuproptosis-associated ncRNAs predict breast cancer subtypes

**Qing Xia[1,2], Jinze Shen[2], Qurui Wang[2], Ruixiu Chen[2], Xinying Zheng[1,2], Qibin Yan[1,2], Lihua Du[2], Hanbing Li[1]\*, Shiwei Duan[2]\***

1 College of Pharmacy, Zhejiang University of Technology, Hangzhou, Zhejiang, China, 2 Key Laboratory of Novel Targets and Drug Study for Neural Repair of Zhejiang Province, School of Medicine, Hangzhou City University, Hangzhou, Zhejiang, China

\* hanbing.li@163.com (HL); duansw@hzcu.edu.cn (SD)

## Abstract

### Background

Cuproptosis is a novel copper-dependent mode of cell death that has recently been discovered. The relationship between Cuproptosis-related ncRNAs and breast cancer subtypes, however, remains to be studied.

### Methods

The aim of this study was to construct a breast cancer subtype prediction model associated with Cuproptosis. This model could be used to determine the subtype of breast cancer patients. To achieve this aim, 21 Cuproptosis-related genes were obtained from published articles and correlation analysis was performed with ncRNAs differentially expressed in breast cancer. Random forest algorithms were subsequently utilized to select important ncRNAs and build breast cancer subtype prediction models.

### Results

A total of 94 ncRNAs significantly associated with Cuproptosis were obtained and the top five essential features were chosen to build a predictive model. These five biomarkers were differentially expressed in the five breast cancer subtypes and were closely associated with immune infiltration, RNA modification, and angiogenesis.

### Conclusion

The random forest model constructed based on Cuproptosis-related ncRNAs was able to accurately predict breast cancer subtypes, providing a new direction for the study of clinical therapeutic targets.

**Data Availability Statement:** All relevant data are within the manuscript and its Supporting Information files. All datasets are available from the TCGA database (https://portal.gdc.cancer.gov/projects/TCGA-BRCA).

**Funding:** This study was supported by the Qiantang Scholars Fund in Hangzhou City University (No. 210000-581835). The funders had no role in study design, data collection and analysis, decision to publish, or preparation of the manuscript.

**Competing interests:** The authors have declared that no competing interest exists.

**Abbreviations:** AUC, Area under the curve; BP, Biological Process; BRCA, Breast invasive carcinoma; CC, Cellular Component; MF, Molecular Function; mRNA, messenger RNA; m1A, N1-methyladensoine; m5C, 5-methylcytosine; m6A, N6-methyladenosine; ncRNA, non-coding RNA; RF, Random Forest; ROC, receiver operating characteristic; ssGSEA, Single-sample gene set enrichment analysis; TCGA, The Cancer Genome Atlas.

## Introduction

According to the World Health Organization, in 2020 there were 2.26 million new cases of breast cancer worldwide [1]. Based on projections by the American Cancer Society, it is anticipated that there will be approximately 288,000 new cases of breast cancer diagnosed in the United States in 2023, with an estimated 43,000 individuals succumbing to the disease [2]. Breast cancer is now the most common malignant tumor in women globally [3]. According to the statistics provided by the Global Cancer Survival Trends Monitor, the five-year survival rate for breast cancer falls within the range of 80–84% [4], with a recurrence rate of 16.6% within 10–32 years following the initial diagnosis [5]. There are many subtypes of breast cancer, and treatment efficacy varies greatly between them. As such, treatment strategies must be tailored to the specific subtype [6]. The PAM50 technology can detect the expression levels of 55 genes and classify breast cancer into 5 subtypes: Luminal A (LumA), Luminal B (LumB), HER2-enriched (Her2), Basal cell type (Basal), and Normal breast cancer-like (Normal) [7, 8]. However, PAM50 detection technology is demanding and expensive [9], so there is a need for new, cheaper alternative detection methods for breast cancer subtypes.

Cuproptosis is a recently discovered copper-dependent mode of cell death [10]. It occurs when excess copper binds directly to fatty acylated components in the tricarboxylic acid (TCA) cycle. This results in fatty acylated protein aggregation and loss of Fe-S cluster proteins, leading to proteotoxic stress and ultimately cell death [10]. High doses of copper can also induce apoptosis by activating a caspase-dependent pathway [11]. Copper ions play a significant role in cancer cell proliferation, metastasis, and angiogenesis during the development of cancer [12, 13]. The copper chelator elesclomol can specifically enter the mitochondria of metabolically active cancer cells, disrupting cellular energy metabolism, inducing oxidative stress, and triggering cancer cell death [14]. Prior investigations have demonstrated that a constructive breast cancer prognosis prediction model can be developed by scrutinizing the involvement of cuproptosis-related genes in breast cancer prognosis. This entails a thorough examination of the correlation between cuproptosis-related genes and crucial factors such as tumor microenvironment and clinical characteristics [15, 16]. Furthermore, the predictive utility of cuproptosis-related genes extends to forecasting drug sensitivity in triple-negative breast cancer patients [17, 18]. Notably, cuproptosis has been identified as a contributor to c-Myc-mediated breast cancer stemness, as highlighted in a recent study [19].

ncRNAs are a class of abundant non-protein-coding RNAs. The classification of ncRNAs based on function and nucleotide length (nt) has been refined as follows: ncRNAs with a length less than 200 nt, such as microRNA (miRNA), transfer RNA (tRNA), and small interfering RNA (siRNA). ncRNAs with a length exceeding 200 nt, encompassing Ribosomal RNA (rRNA), long non-coding RNA (lncRNA), and circular RNA (circRNA) [20]. ncRNAs can regulate the expression of various cancer-related genes and participate in numerous biological regulatory processes, including tumor occurrence, development, and metastasis [21]. As the biological functions of ncRNAs continue to be explored, it has been discovered that they represent a higher level of gene regulation. A single ncRNA could theoretically control the expression of many downstream mRNAs [22].

This study aims to establish a breast cancer subtype prediction model based on Cuproptosis-related ncRNAs using a random forest algorithm. By identifying biomarkers of breast cancer subtypes, this study is expected to provide a new direction for the investigation of clinical therapeutic targets.

## Materials and methods

### Biological function analysis of Cuproptosis-related genes

As shown in Fig 1, we downloaded RNA-seq data of breast cancer and paracancerous tissues from the TCGA database (https://portal.gdc.cancer.gov/). We then obtained the subtype classification [23] of TCGA-BRCA patients (S1 Table). After excluding patients with unclear subtypes, we had a total of 1106 tumor samples and 101 paracancerous samples. These 1106 patients were randomly divided into training (776 patients) and testing (330 patients) groups using the R statistical software caret package [24] (Table 1). We identified Cuproptosis-related genes through literature analysis and used the R statistical software "clusterProfiler" package [25] to analyze their biological functions and pathways.

### Identification of Cuproptosis-associated differential ncRNAs

We used the voom algorithm of the limma package [26] to identify differentially expressed ncRNAs in breast cancer tissues and paracancerous tissues. Significant differential expression was set as false discovery rate (FDR) <0.05 and an absolute value of log fold-change (($|logFC|$) $\geq$2. Pearson correlation analysis was used to evaluate the relationship between differentially expressed ncRNAs and Cuproptosis-related genes. The association was considered significant if the absolute value of correlation coefficient ($|R|$)>0.3 and FDR<0.001.

### Establishment and evaluation of breast cancer subtype prediction model

A breast cancer subtype prediction model was established and evaluated using the Random Forest (RF) algorithm from the randomForest package for feature selection and model building [27]. The dimensionality of features was reduced using a random survival forest algorithm, and gene selection and model building were performed based on variable importance (VIMP) and minimum depth. The SHAP package was used to interpret the machine learning models [28], providing the degree of influence of each feature in the model and its positive or negative impact on each prediction result.

### Analysis of immune infiltration based on Cuproptosis-related ncRNAs

The tumor immune microenvironment consists primarily of tumor cells, fibroblasts, immune cells, and an array of signaling molecules. It is well-established that the tumor microenvironment plays a pivotal role in influencing tumor diagnosis, survival outcomes, and clinical

**Table 1. Sample collection and grouping of TCGA-BRCA dataset.**

| Subtype | Training group (n) | Testing group (n) |
|---|---|---|
| Basal | 124 | 53 |
| Her2 | 52 | 21 |
| LumA | 362 | 155 |
| LumB | 135 | 57 |
| Normal | 103 | 44 |
| Total | 776 | 330 |

We obtained 1106 BRCA samples from the TCGA database and classified the subtypes of TCGA-BRCA patients based on the literature. The patients were then randomly divided into a training group (n = 776) and a test group (n = 330) at a ratio of 7:3.

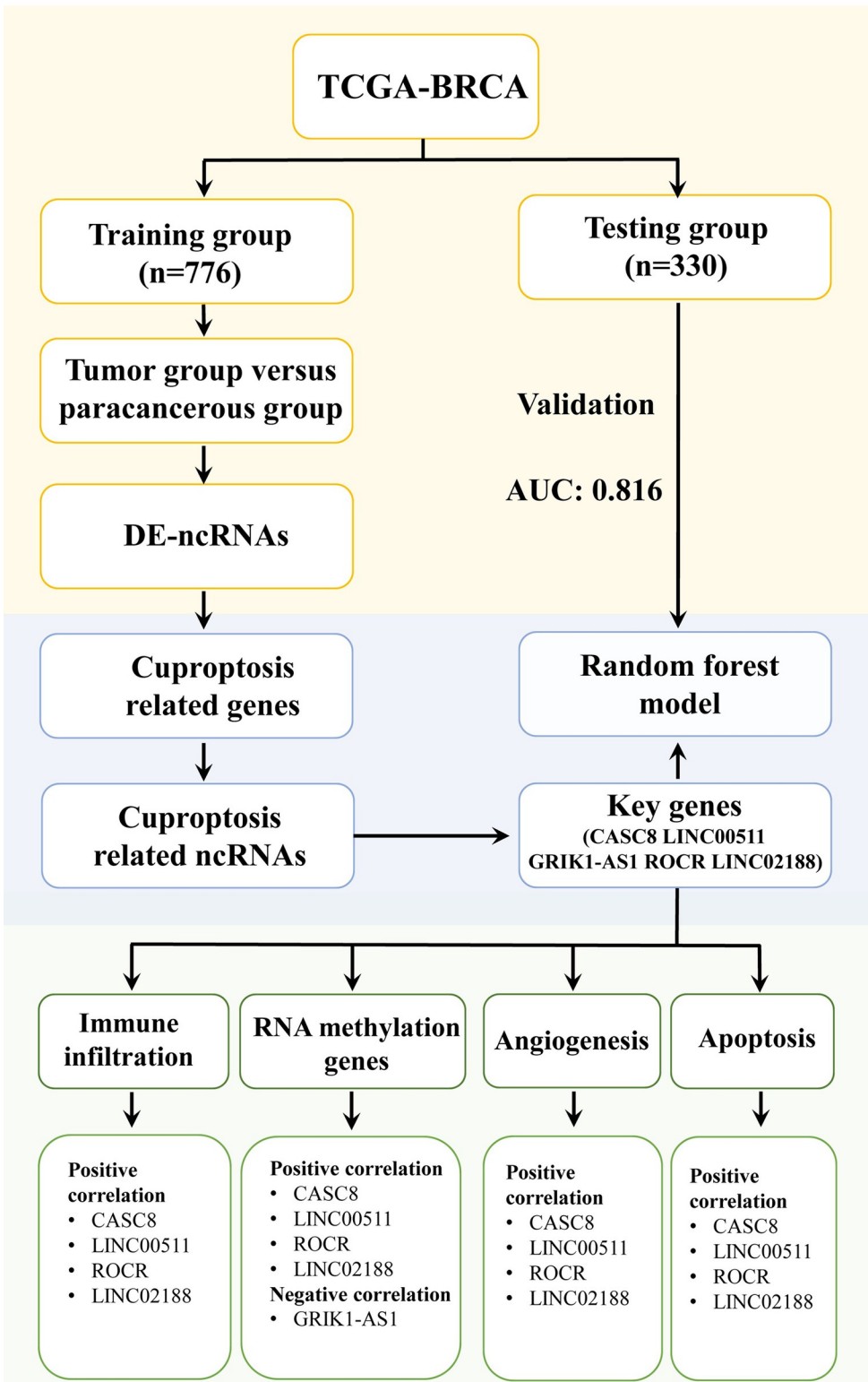

**Fig 1. The flow diagram of data collection and analysis in the present study.** In this study, we screened for differentially expressed ncRNAs using the TCGA-BRCA dataset. We then combined this with Cuproptosis-related gene screening to identify key ncRNAs. A random forest model was established to predict the function of these key ncRNAs.

treatment responsiveness [29]. Thus, gaining insights into the composition of immune cells within tumors is indispensable for advancing cancer treatment strategies.

Immune infiltration was analyzed based on Cuproptosis-related ncRNAs using the CIBER-SORT (https://cibersort.stanford.edu/) deconvolution algorithm to calculate tumor immune cell composition from expression profiles [30]. Single-sample gene set enrichment analysis (ssGSEA) from the GSVA package [31] was used to calculate the degree of infiltration of 28 immune cell types based on published immune cell gene sets [32]. The correlation between Cuproptosis-related ncRNAs and immune infiltration was calculated to explore their relationship in different breast cancer subtypes.

Immune checkpoints play a pivotal role as crucial molecules employed by the immune system to finely tune the expression of its own proteins. Tumor cells, adept at evading immune responses, often achieve this evasion through the intricate regulation of immune checkpoints [33]. To delve into the association between cuproptosis-related ncRNAs and immune checkpoints, we meticulously gathered data on 20 immune checkpoints from the TISIB website. Employing Pearson correlation analysis, we sought to unravel the nuanced relationship between these ncRNAs and the intricate network of immune checkpoints.

## Analysis of RNA modifications based on Cuproptosis-related ncRNAs

RNA methylation, a post-transcriptional modification that regulates multiple biological functions and is associated with tumor development and malignant progression, affects gene expression by regulating RNA metabolism, splicing, stability, and translation [34]. Three RNA modification-related genes were obtained from the literature, including 23 m6A modification genes [35], 12 m5C modification genes [36], and 10 m1A modification genes [37] (S2 Table). The correlation between Cuproptosis-related ncRNAs and RNA modifying genes was calculated to explore their relationship in different breast cancer subtypes.

## Analysis of angiogenesis based on Cuproptosis-related ncRNAs

Angiogenesis, the growth of new capillaries from pre-existing vessels, is essential for tumor development and metastasis [38]. 36 angiogenesis-related mRNAs were obtained from the MSigDB team's Hallmark gene set [39] (S3 Table). The association between Cuproptosis-related ncRNAs and angiogenesis-associated genes (*AAGs*) was assessed to explore their relationship in different breast cancer subtypes.

## Analysis of apoptosis based on Cuproptosis-related ncRNAs

We compiled a set of 20 apoptosis-related protein-coding genes from relevant literature sources. Subsequently, we conducted a comprehensive analysis to elucidate the correlation between cuproptosis-related ncRNAs and the 20 apoptotic genes. Our investigation sought to uncover the distinctive roles played by these molecular components in various subtypes of breast cancer.

## Statistical analysis

All statistical analyses were performed using R version 4.1.0 (http://www.r-project.org/) (packages including limma, pROC, caret, Random Forest, ggplot and GSVA). Gene expression was compared between tumor tissue and adjacent non-tumor tissue using a t-test. Pearson analysis was used to calculate correlations between genes, and Wilcoxon tests were used to compare differences in proportions between groups. The sensitivity and specificity of the models were assessed using receiver operating characteristic (ROC) curves and area under the

curve (AUC). All reported p-values correspond to two-sided tests, with p-values <0.05 considered statistically significant.

### Ethics statement and informed consent statement

The Cancer Genome Atlas (TCGA) is classified as a public database, housing data from patients who have received ethical approvals for inclusion. Within this database, users have the privilege of freely accessing relevant data for research and the subsequent publication of related articles. Our study exclusively relies on open-source data, which inherently mitigates any ethical concerns or potential conflicts of interest. It is imperative to highlight that informed consent was duly obtained from all participants involved in our study, underscoring the ethical rigor of our research.

## Results

### Enrichment analysis of Cuproptosis-related genes

We conducted an enrichment analysis of Cuproptosis-related genes by identifying 21 Cuproptosis-related mRNAs through a literature search (Fig 2). Their enriched pathways and involved molecular functions were analyzed. Gene Ontology results showed that these genes are involved in biological processes that respond to metal ions and regulate transition metal ion homeostasis. They are located in the mitochondrial matrix of cells and have the primary molecular function of binding copper ions. KEGG pathway analysis revealed that most of the genes were mainly involved in the platinum resistance pathway, p53 signaling pathway, and cell apoptosis pathway (S1 Fig and S3 Table). Furthermore, our analysis, utilizing follow-up data from the TCGA database, delved into the survival rates across five distinct subtypes: Basal, Her2, Lum A, Lum B, and Normal. Significantly divergent survival rates were observed among these subtypes (P<0.0001). Notably, the Lum A subtype exhibited the highest five-year survival rate, reaching an impressive 89.5%. Conversely, the Normal subtype presented the lowest five-year survival rate, standing at only 72% (Fig 3A).

We identified the expression of cuproptosis-related mRNA across five distinct breast cancer subtypes, revealing that only two genes, PDHB and PRNP, exhibited significant differential expression within this cohort (S2 Fig). Employing cuprotosis-associated genes, we constructed a breast cancer subtype prediction model, unveiling an area under the curve (AUC) of 0.74 in the validation set. This result suggests suboptimal performance for the prediction model relying on cuprotosis-related genes. Subsequently, we endeavored to identify cuproptosis-related ncRNAs and establish a prediction model with superior performance.

### Feature selection

We screened for differentially expressed ncRNAs related to Cuproptosis by performing a differential analysis on the TCGA cohort training set, which included 730 tumor samples and 101 normal samples. A total of 194 differentially expressed ncRNAs were found in breast cancer, with 52 up-regulated and 142 down-regulated (Fig 3B, S4 Table). Pearson correlation analysis was then used to identify 94 ncRNAs that were significantly correlated with Cuproptosis (S5 Table).

We use random forest algorithm to rank the importance of 94 ncRNAs related to Cuproptosis. The five genes with the highest relative importance value were selected as key genes and included in the construction of the final model (Fig 3C and 3D).

The expression levels of the five key genes used to build the model were significantly different among the five subtypes (Fig 3E–3G, S6 Table). In the Basal subtype, *CASC8* and

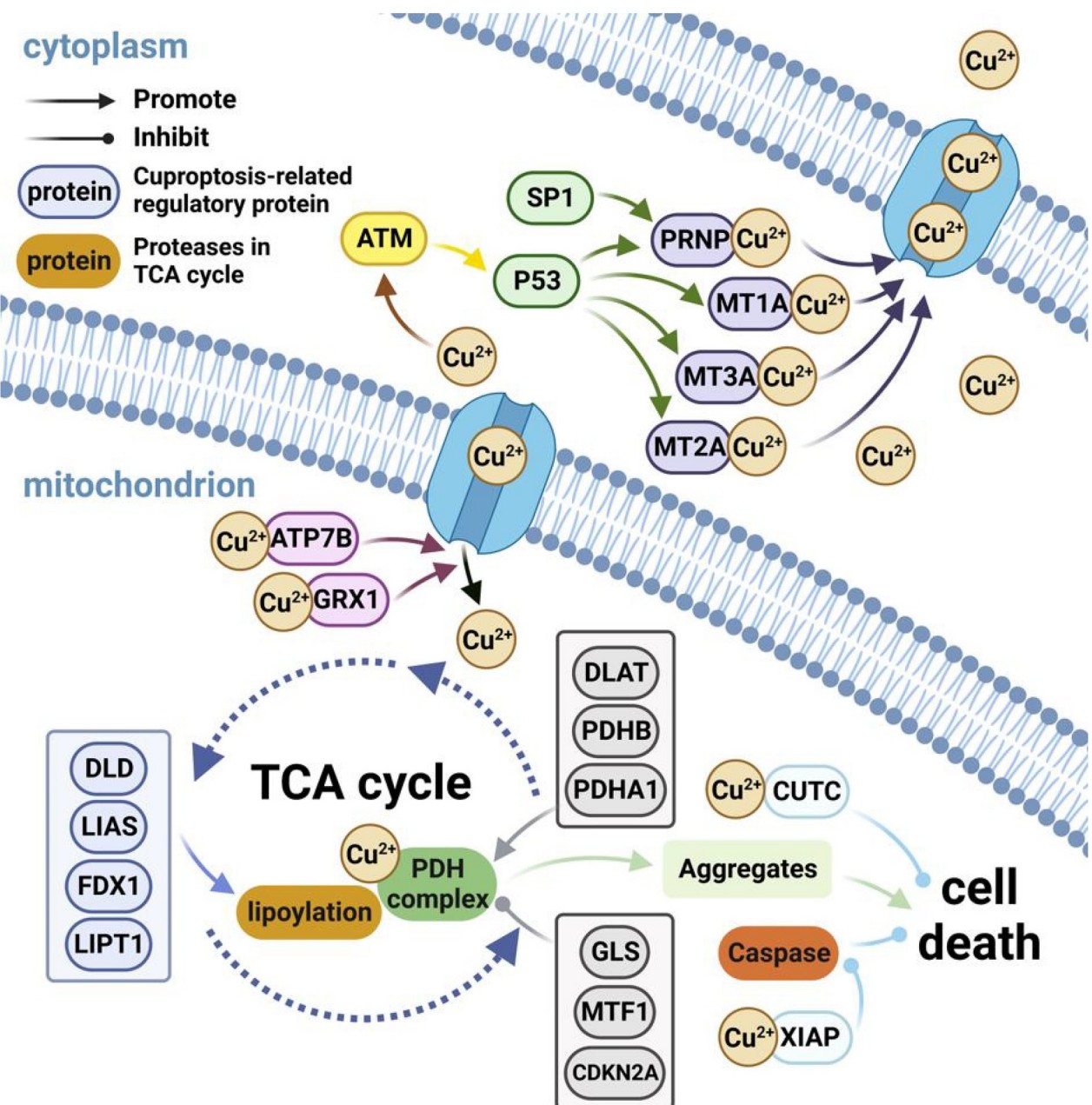

**Fig 2. Molecular mechanism of Cuproptosis-related genes.** We retrieved 21 genes involved in Cuproptosis from the literature, including CDKN2A, DLD, DLAT, FDX1, GLS, LIAS, LIPT1, MTF1, PDHA1, and PDHB, ATB7B, CUTC, GRX1, MT1A, MT2A, MT3, XIAP, ATM, PRNP, P53, and SP1. We showed the mechanism of action of these genes in Cuproptosis. Excess $Cu^{2+}$ stimulates the activation of ATM which activates SP1 and P53. P53 promotes PRNP, MT1A, MT2A and MT3A to expel $Cu^{2+}$ out of the cell. Four genes including DLD encode lipoylation-related enzymes. Three genes including DLAT positively regulate the PDH complex in the TCA cycle while three genes including GLS negatively regulate it. The lipoylated PDH complex combined with $Cu^{2+}$ leads to oligomerization and aggregate formation which triggers cell death. ATP7B and GRX1 excrete excess copper ions out of the mitochondria. CUTC directly binds $Cu^{2+}$ while XIAP passivates and hydrolyzes caspase after binding $Cu^{2+}$ to inhibit Cuproptosis.

*LINC02188* had the highest expression (0.75–1 quantile, Q4), while *GRIK1-AS1* had the lowest (0.25–0.5 quantile, Q2). In the Her2 subtype, *LINC00511* had the highest expression (Q4), while *ROCR* had the lowest (Q2). In LumA subtypes, *GRIK1-AS1* had the highest expression (Q4), while *LINC00511* had the lowest (0.5–0.75 quantile, Q3). In LumB subtypes, *LINC00511*

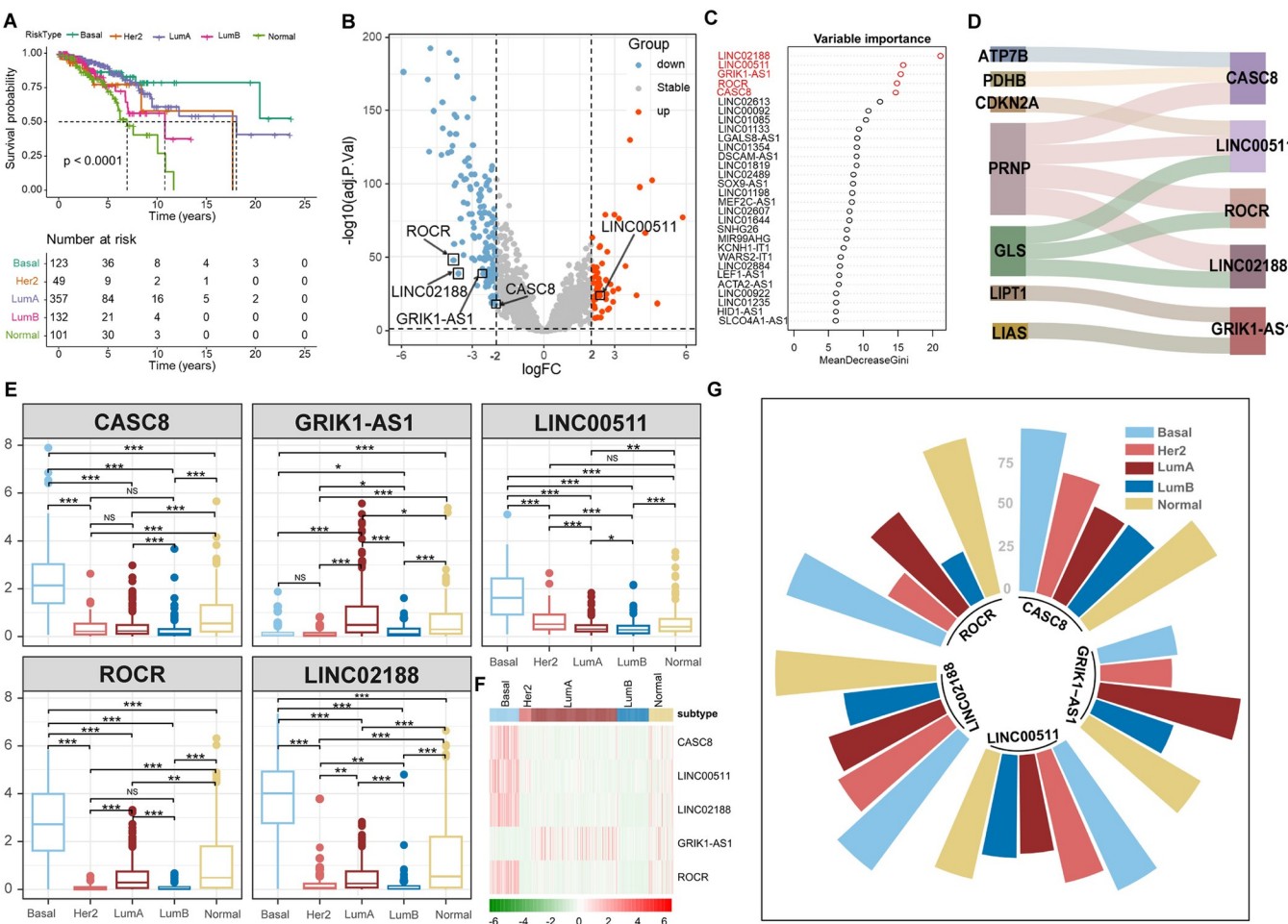

**Fig 3. Expression of key ncRNAs in breast cancer subtypes.** (A) Survival analysis of the five subtypes. (B) Key genes are differentially expressed between breast cancer and paracancerous cells. Red indicates down-regulated genes while blue indicates up-regulated genes. (C) Five genes with MeanDecreaseGini above 5 are selected from the top 20 features to build a model. (D) Cuproptosis-related genes show significant correlation with key genes. The thickness of the connecting lines in the figure represents the absolute value of Pearson's R (p<0.05). (E) The box plot displays the significant differential expression of key genes across different subtypes. * indicates P<0.01, ** indicates P<0.001 and *** indicates P<0.0001. (F) A heatmap shows the expression of key ncRNAs in different subtypes. (G) The height of the bars represents the expression quantile of key ncRNAs. We calculated the median expression of all ncRNAs in the 5 breast cancer subtypes and determined the rank percentage of each key lncRNA among all non-zero expressed ncRNAs.

had the highest expression (Q3), while *ROCR* had the lowest (Q2). In the Normal subtype, all five genes have high expression. (Q4).

## Evaluation of the random forest predictive models

The breast cancer subtype prediction model was constructed using the random forest algorithm based on five key genes. The results demonstrated an area under the curve of 0.816 and an F1 Score of 0.57 for the validation set. Additionally, precision was determined to be 0.56, specificity (Sp) as 0.81, and sensitivity (Sn) as 0.61. These indicating that the random forest model can accurately predict breast cancer subtypes (Fig 4A and 4B).

We evaluated the breast cancer subtype risk factor model built using the SHAP library (Fig 4C). The random forest model predicted the Basal subtype in a patient with an accuracy of 0.94, with *ROCR*, *GRIK1-AS1*, *LINC02188*, CASC8 and *LINC00511* increasing the prediction accuracy. The model predicted the Her2 subtype in another patient with an accuracy of 0.16,

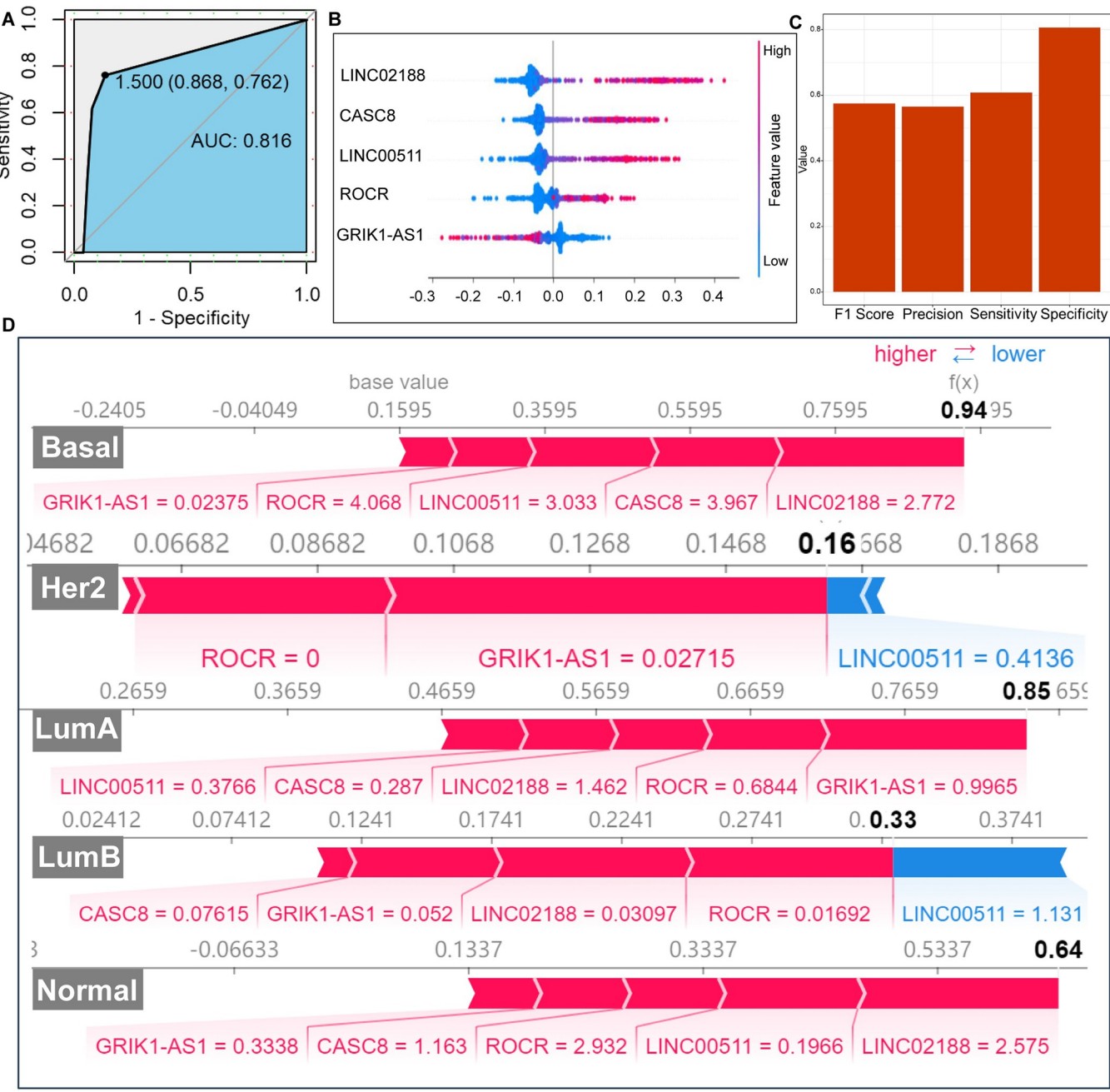

**Fig 4. Establishment and validation of models related to Cuproptosis.** (A) The ROC curve of the verification set has an AUC value of 0.816. (B) F1 Score, Precision, Sensitivity (Sn) and Specificity (Sp) of verification set. (C) The SHAP Disease Risk Factor Model identifies the most important features and their range of effects over the dataset. The color represents the feature value with red indicating high values and blue indicating low values. (D) Detailed explanations of model predictions for 5 individual samples. Red values indicate factors that positively impact subtype prediction while blue values indicate factors that negatively impact subtype prediction. The longer the arrow, the greater the influence of the feature on the output.

with *GRIK1-AS1* increasing the prediction accuracy and *LNC00511* decreasing it. The model predicted the LumB subtype in a fourth patient with an accuracy of 0.33, with *GRIK1-AS1*, *LNC00511*, *LINC02188* and *CASC8* increasing the prediction accuracy and *LNC00511* decreasing it. The model predicted the LumA subtype in a third patient with an accuracy of 0.85 and the Normal subtype in a fifth patient with an accuracy of 0.64. The above results showed that

five key genes—*CASC8*, *LINC02188*, *LINC00511*, *GRIK1-AS1*, and *ROCR*—had varying contributions in predicting breast cancer subtypes. (Fig 4D)

## Immune infiltration and the key ncRNAs

We conducted an immune infiltration assay to explore the relationship between Cuproptosis and immune infiltration in different breast cancer subtypes. CIBERSORT and ssGSEA were used to evaluate the immune infiltration of patients, and the correlation between key genes and immune cells was calculated. Key genes were found to have varying degrees of correlation with the level of immune cell infiltration (Fig 5A, S7 Table). T cells, activated dendritic cells, natural killer cells, plasmacytoid dendritic cells, and M1 macrophages were significantly negatively correlated with *GRIK1-AS1* and significantly positively correlated with *CASC8*, *ROCR*, *LINC00511* and *LINC02188*. M2 macrophages was significantly negatively correlated with *CASC8*, *ROCR*, *LINC00511* and *LINC02188*. Mast cells were significantly positively correlated with *GRIK1-AS1*.

Plasmacytoid dendritic cells promote innate and adaptive immune responses by inducing the migration of natural killer (NK) cells, the maturation of dendritic cells, and the differentiation of T cells [40]. Macrophages are commonly divided into M1 and M2 subtypes [41], with M1 being pro-inflammatory and anti-tumor and M2 being anti-inflammatory and tumor-promoting. T cells play a key role in tumor surveillance by recognizing and eliminating transformed cells [42]. Mast cells are typically located at the edge of tumors and stimulate angiogenesis, extracellular matrix breakdown, and tumor growth by releasing mediators such as *VEGF* and *MMP9* [43].

Since *GRIK1-AS1* were most expressed in the LumA subtype and *LINC02188* and *ROCR* were most expressed in the Basal and Normal subtypes, we speculate that high expression of *GRIK1-AS1* in the LumA subtype may promote tumor development by activating mast cells. In contrast, high expression of *LINC02188* and *ROCR* in the Basal and Normal subtypes may increase tissue tumor infiltration and inhibit tumor development by activating T cells, plasmacytoid dendritic cells, NK cells, M1 macrophages, and other immune cell activities.

To investigate the intricate connection between cuproptosis-associated ncRNAs and the phenomenon of immune escape, we conducted a comprehensive correlation analysis to assess the interrelation of key genes with immune checkpoints. The results, depicted in Fig 5B, reveal a positive correlation between LINC0511, LINC02188, and ROCR, and eight pivotal immune checkpoint genes, namely CD96, CTLA4, LAG3, PD-1, PD-L2, PD-L1, TIGIT, and VTCN1. Furthermore, our investigation unearthed that LINC02188 exhibits the highest expression levels within the Basal subtype, implying a heightened vulnerability to immune escape among patients characterized by the Basal subtype, as substantiated by prior research findings.

## Angiogenic genes and the key ncRNAs

Angiogenesis, the formation of new blood vessels, has been shown to be integral to cancer development [44]. Our study assessed the correlation of key genes with angiogenic genes to explore the relationship between Cuproptosis and angiogenesis in different breast cancer subtypes. The results found that *CASC8*, *ROCR* and *LINC02188* were significantly positively correlated with TNFRSF21, PRG2, APP and CXCL6 (R>0.24, P<0.0001). *CASC8*, *ROCR* and *LINC02188* were significantly negatively correlated with SERPINA5, POSTN, COL5A2, COL5A1 (|R|>0.17, P<0.0001). *LINC02188* had the highest expression in the Basal and Normal subtype, suggesting that this subtype may have the highest angiogenesis. We speculated that specifically reduced the expression of *LINC02188* may improve angiogenesis in the Basal and Normal subtype (Fig 5C, S8 Table).

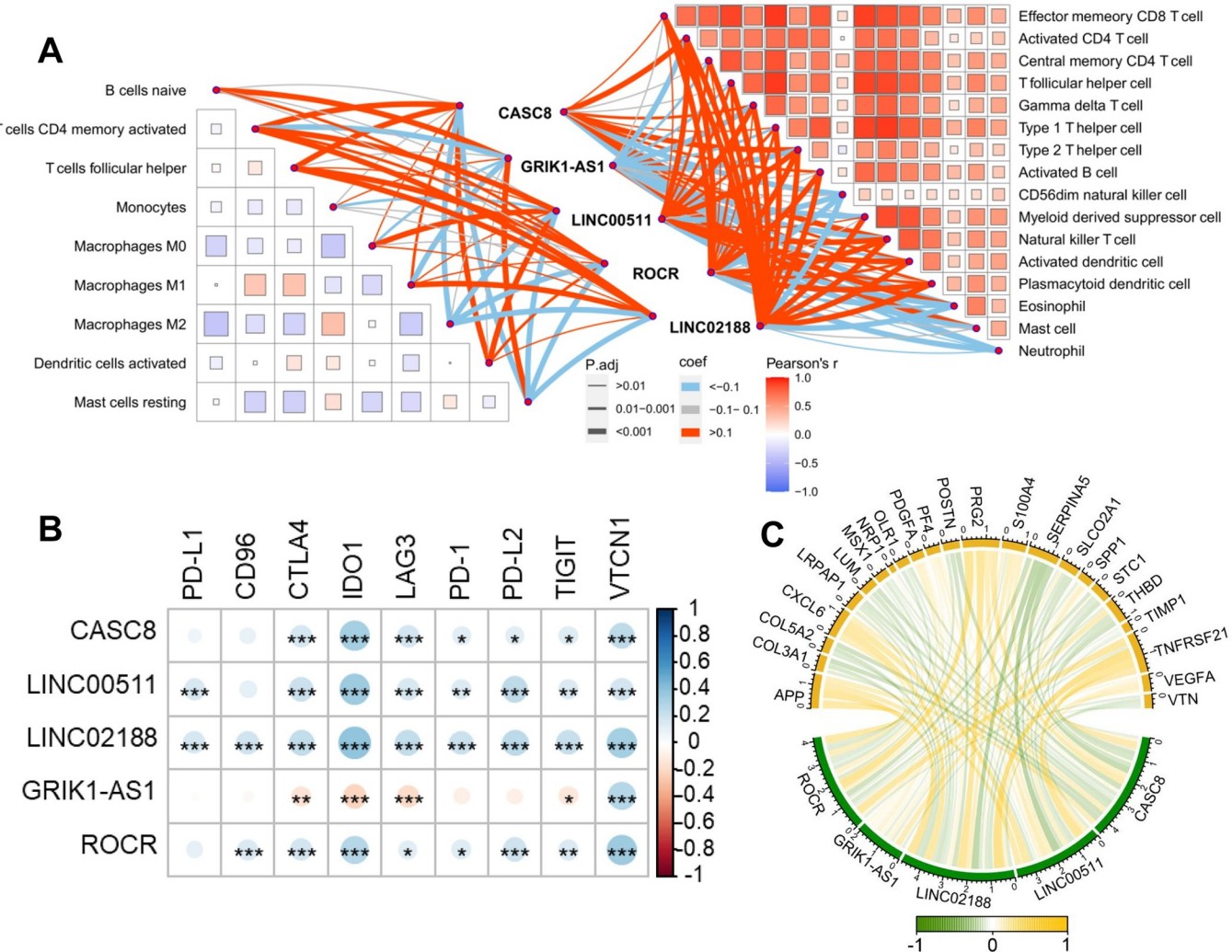

**Fig 5. Key ncRNAs in breast cancer subtypes associated with immune infiltration, RNA methylation modification, and angiogenesis.** (A) Right: ssGSEA is used to assess the enrichment scores of immune pathways in breast cancer patients. Correlations exist between immune pathways and between Key ncRNAs in breast cancer subtypes and immune pathways. Left: CIBERSORT is used to assess the proportion of immune cells in breast cancer patients. Correlations exist between immune cells and between Key ncRNAs in breast cancer subtypes and immune cells. Pearson's r represents the correlation coefficient between immune cells while coef represents the correlation coefficient between Key ncRNAs in breast cancer subtypes and immune cells. P.adj represents the adjusted significance P value of Key ncRNAs in breast cancer subtypes and immune cells. (B). Key ncRNAs in breast cancer subtypes show correlation with immune checkpoints. (C) Key ncRNAs in breast cancer subtypes show correlation with angiogenesis-related genes. Yellow represents positive correlation while green represents negative correlation. The thickness of the line represents the absolute value of the correlation coefficient.

## Apoptosis genes and the key ncRNAs

This study delved into the intricate association between cuproptosis and apoptosis across various subtypes of breast cancer cells. The investigation centered on evaluating the correlation between key cuproptosis-related ncRNAs and apoptosis-related genes. The findings illuminated a noteworthy negative correlation between CASC8, ROCR, and LINC02188 with 10 apoptosis genes, as depicted in Fig 6A. Particularly, CASC8 and LINC02188 exhibited the highest expression levels in the Basal and Normal subtypes. This observation suggests a compelling inference that CASC8 and LINC02188 may exert inhibitory effects on apoptosis by suppressing the expression of apoptosis-related proteins within the Basal and Normal subtypes.

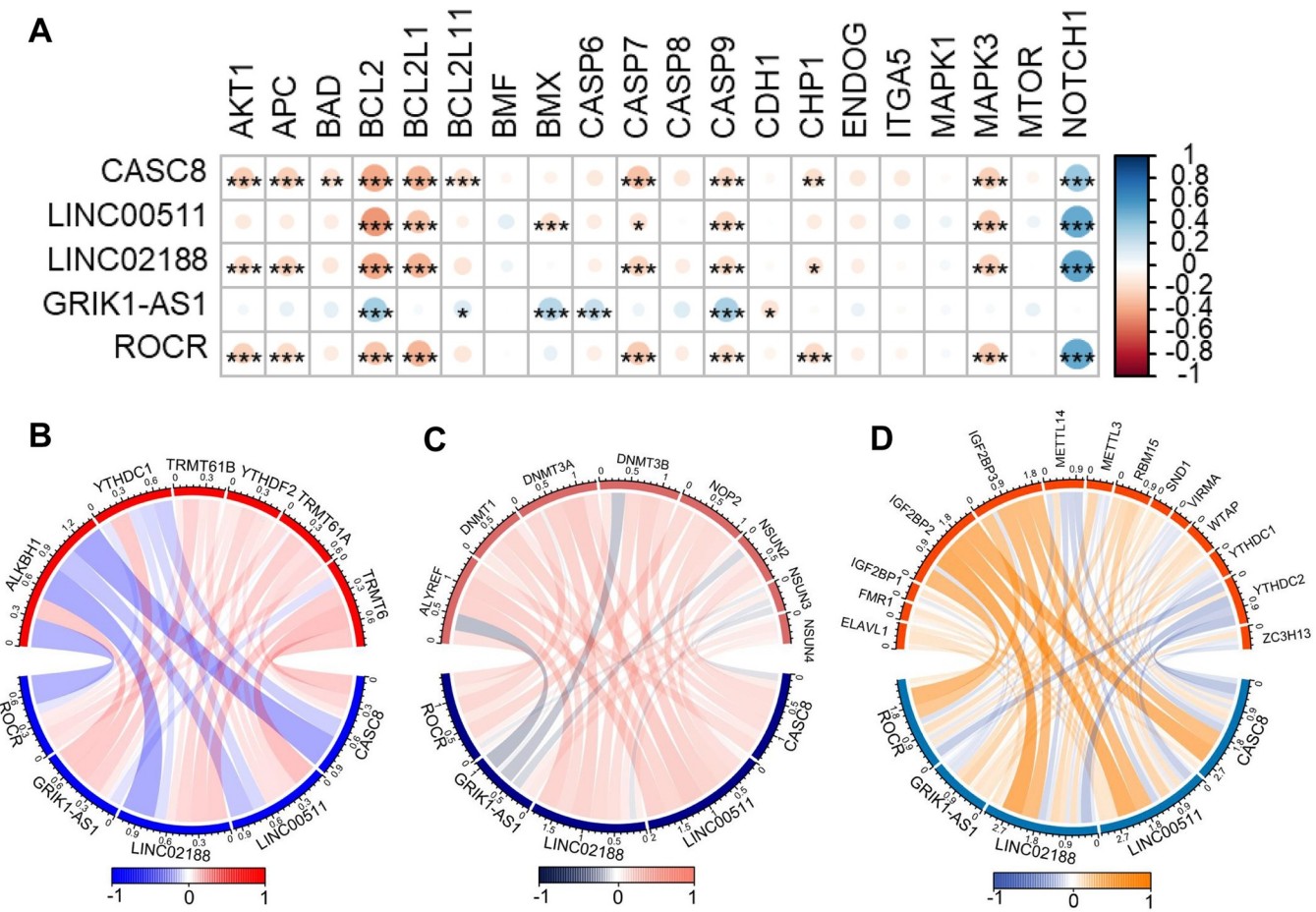

**Fig 6. Key ncRNAs in breast cancer subtypes associated with apoptosis and RNA methylation modification.** (A). Key ncRNAs in breast cancer subtypes show correlation with apoptosis-related genes. (B) Key ncRNAs in breast cancer subtypes show correlation with m1A modification-related genes. Red represents positive correlation while blue represents negative correlation. (C) Key ncRNAs in breast cancer subtypes show correlation with m5C modification-related genes. Red represents positive correlation while darkblue represents negative correlation. (D) Key ncRNAs in breast cancer subtypes show correlation with m6A modification-related genes. Orange represents positive correlation while blue represents negative correlation.

### RNA methylation genes and the key ncRNAs

We explored the relationship between Cuproptosis and RNA methylation in different breast cancer subtypes by assessing the correlation of key genes with RNA methylation genes. RNA methylation plays a crucial role in cancer development. For example, m1A modification encompasses a range of critical effects on RNA, influencing processes such as RNA processing, secondary and tertiary structure, stability, translation efficiency, and biological function [45]. A pivotal contributor to m1A modification is *TRMT61B*, a mitochondria-specific tRNA methyltransferase primarily responsible for installing m1A at position 58 (m1A58) of tRNA [46]. Additionally, *TRMT6* and *TRMT61A* have been implicated in glioma, gastrointestinal cancer, and hepatocellular carcinoma (HCC) development [47]. Notably, *ALKBH1*, exhibiting demethylase activity, catalyzes the demethylation of m1A modification [48]. *CASC8*, *LINC00511* and *LINC02188* were significantly positively correlated with m1A modifier genes (*TRMT61B*, *YTHDF2*, *TRMT61A*, *TRMT6*) and significantly negatively correlated with *ALKBH1*. (Fig 6B, S8 Table)

The m5C modification, on the other hand, is intimately involved in the progression of various tumors. NOP2/Sun domain family members 1–7 (*NSUM 1–7*) and the *DNMT* homolog

*DNMT2* serve as m5C writers in mammals, orchestrating the methylation of the RNA C5 position [49]. Conversely, *TET2* acts on m5C, converting it into 5-hmC and eventually removing the methyl group [50]. Furthermore, *ALYREF* functions as a reader, specifically recognizing and binding to the m5C motif, thereby exerting additional biological functions [51]. *CASC8, LINC02188, ROCR and LINC00511* was significantly positively correlated with m5C modifier genes (Fig 6C, S8 Table).

Finally, m6A modification plays a vital role in governing mRNA stability, splicing, degradation, and translation efficiency [52]. The generation of m6A modification hinges on the methyltransferase complex, consisting of METTL3/METTL14/WTAP proteins. Concurrently, *YTHDC1/2*, *YTHDF1/2/3*, *IGF2BP1/2/3*, and *HNRNP* act as readers, proficient in recognizing m6A residues [53]. *CASC8*, *LINC00511*, *LINC02188* and *ROCR* was significantly positively correlated with *IGF2BP2*, *IGFBP3*, *RBM15*, *SND1*, *WTAP* (Fig 6D, S8 Table).

Cuproptosis has a complex relationship with RNA methylation, and reducing the level of RNA methylation can inhibit tumor development. It is therefore reasonable to hypothesize that targeting *LINC02188* and *ROCR* in the Basal subtype and targeting LINC00511 in the Her2 and LumB subtype could reduce RNA methylation modification levels to inhibit tumor development. However, these hypotheses require further validation in the laboratory. The results suggest that key genes may be promising clinical therapeutic targets.

It is well-established that the development of new blood vessels plays a pivotal role in the progression of solid tumors, serving as a crucial precursor for tumor invasion and metastasis [54]. Recent investigations have revealed that IGF2BP2 has the capability to infiltrate endothelial cells via exosomes, thereby fostering angiogenesis and metastasis in LUAD [55]. METTL14, through the modulation of TGFβ via the RhoA and PI3K-AKT pathways, contributes to tumor angiogenesis and progression [56]. Additionally, YTHDF3 has been identified as a promoter of cancer cell-astrocyte interaction, thereby facilitating angiogenesis. Notably, current research underscores the therapeutic potential of targeting METTL3 to combat angiogenesis in bladder cancer [57].

Furthermore, tumor neovascularization is implicated in the secretion of angiogenic factors that suppress the activity of cytotoxic T cells, culminating in tumor immunosuppression [58]. Concurrently, tumor cells or macrophages within the tumor microenvironment release substantial quantities of pro-angiogenic factors such as VEGF, bFGF, and PDGF. These factors activate pro-angiogenic signaling pathways, fostering the growth, invasion, and metastasis of tumor blood vessels [59]. Collectively, these studies underscore the role of RNA methylation and the tumor microenvironment in promoting tumor angiogenesis, presenting them as potential targets for cancer treatment.

As indicated in Table 2, this study identified high expression of LINC02188 in the Basal subtype and elevated expression of GRIK1-AS1 in LumA. Notably, five cuproptosis-related ncRNAs demonstrated a significant positive correlation with RNA methylation, angiogenesis factors, and immune cells. This correlation suggests that in the Basal subtype, LINC02188 may enhance the release of tumor angiogenesis factors (TNFRSF21, PRG2) by augmenting RNA methylation, consequently promoting angiogenesis. Similarly, in the LumA subtype, GRI-K1-AS1 might mitigate cancer malignancy by inhibiting T cell and B cell activity, thereby reducing the extent of RNA methylation modification.

## Discussion

Cuproptosis is a recently discovered type of cell death that is distinct from apoptosis, autophagy, and ferroptosis. In this study, we employed a random forest model to discern five breast cancer subtype biomarkers linked to Cuproptosis. Among these, four ncRNAs (*CASC8*,

**Table 2.** Associated changes in key genes.

| ncRNAs | | CASC8 | GRIK1-AS1 | LINC00511 | ROCR | LINC02188 |
|---|---|---|---|---|---|---|
| Subtype | Basal | 2.34 ± 1.4 | 0.15 ± 0.2 (lowest) | 1.72 ± 0.9 | 2.71 ± 1.6 | 3.73 ± 1.6 (highest) |
| | Her2 | 0.39 ± 0.5 | 0.12 ± 0.1 | 0.68 ± 0.5 (highest) | 0.08 ± 0.1 (lowest) | 0.31 ± 0.6 |
| | LumA | 0.37 ± 0.4 | 0.86 ± 0.9 (highest) | 0.36 ± 0.2 (lowest) | 0.51 ± 0.6 | 0.5 ± 0.6 |
| | LumB | 0.27 ± 0.4 | 0.22 ± 0.3 | 0.36 ± 0.3 (highest) | 0.08 ± 0.1 (lowest) | 0.16 ± 0.5 |
| | Normal | 0.95 ± 1.1 | 0.69 ± 0.8 | 0.66 ± 0.7 | 1.24 ± 1.6 | 1.4 ± 1.8 (highest) |
| RNA methylation | m1A | Positive correlation | Negative correlation | Positive correlation | Positive correlation | Positive correlation |
| | m5C | Positive correlation | Negative correlation | Positive correlation | Positive correlation | Positive correlation |
| | m6A | Positive correlation | Negative correlation | Positive correlation | Positive correlation | Positive correlation |
| Immune | checkpoint molecules | Positive correlation | Negative correlation | Positive correlation | Positive correlation | Positive correlation |
| | Immune cells (Suppress tumor) | Positive correlation | Negative correlation | Positive correlation | Positive correlation | Positive correlation |
| | Immune cells (Promote tumor) | Negative correlation | Positive correlation | Negative correlation | Negative correlation | Negative correlation |
| angiogenesis | | Positive correlation | - | Positive correlation | Positive correlation | Positive correlation |
| apoptosis | | Negative correlation | - | Negative correlation | Negative correlation | Negative correlation |

*GRIK1-AS1*, *LINC02188*, and *ROCR*) exhibit diminished expression levels in breast cancer, while one lncRNA (*LINC00511*) displays elevated expression in breast cancer. Notably, *CASC8*, *ROCR*, *LINC02188*, and *LINC00511* exhibit increased expression within the Basal and Normal subtypes. Conversely, *GRIK1-AS1* shows elevated expression in the LumA subtype.

Previous studies have underscored the significance of these ncRNAs in breast cancer biology. *ROCR* has been implicated in promoting breast cancer proliferation by facilitating the expression of the oncogenic transcription factor *SOX9* [60]. *CASC8* is associated with the risk of multiple cancers, including non-small cell lung cancer, retinoblastoma, esophageal squamous cell carcinoma, and colorectal cancer [61–63], and has been found to be highly expressed in hepatoma cells like MIA PaCa-2 [64]. *LINC00511* may upregulate the expression of MMP13 by acting as a miR-150 sponge, thereby promoting breast cancer cell proliferation, migration, and invasion [65]. *GRIK1-AS1* serves as a sponge for miR-375, leading to increased *IFIT2* protein levels and the promotion of gastric cancer progression [66]. Additionally, *GRIK1-AS1* can impede *DNMT1* from binding to the *SRFP1* promoter, resulting in *SFRP1* hypomethylation and subsequent upregulation, thereby accelerating the progression of endometriosis [67]. Presently, limited studies have explored the functions of *LINC02188*. These findings underscore the close associations between most of the key genes and the onset and progression of cancer. They assume distinct roles in the development and advancement of tumors within different subtypes. As such, they hold the potential for use as subtype identification markers and are prospective targets for future breast cancer treatments.

ncRNA is closely related to immune cells. Knockdown of lncRNA *KCNQ1OT1* in colorectal cancer cells reduces *CD620* expression and enhances the immune response of CD8 T cells [68]. Knockdown of *MIR155HG* suppresses M2 macrophage polarization and CRC progression in nude mice [69]. We predict that targeted inhibition of *GRIK1-AS1* expression in the LumA subtype can increase the proportion of T cells and promote immune responses, enhancing tumor infiltration and immunological capabilities. In the Her2 and LumB subtypes, promoting the expression of *CASC8*, *ROCR* and *LINC02188* to reduce the proportion of M2 macrophages may make it easier to establish an immune-infiltrating microenvironment [70], increasing the effectiveness of immunotherapy.

We built a breast cancer subtype prediction model and identified biomarkers based on Cuproptosis-associated ncRNAs using a random forest approach on 730 breast cancer patients

from the TCGA training set. The random forest algorithm is an ensemble algorithm with high accuracy and precision, especially for multi-featured data [71]. It has been widely used in recent years and has successfully detected and predicted type 2 diabetes [72] and metabolic syndrome [73]. In this study, the AUC of the ROC curve for the random forest model was 0.816 in the test set, but the accuracy and F1 score of random forest model are low, only 0.56 and 0.57. We hypothesize that too many label classifications and a large sample size gap between subtypes may have led to model overfitting.

Previous research has delved into cell death patterns concerning breast cancer, with a particular focus on copper death. For instance, a study demonstrated that regulators associated with copper-induced cell death can serve as prognostic indicators, predict chemotherapy response, and anticipate immunotherapy outcomes in breast cancer patients [74]. Deng et al. harnessed machine learning techniques to identify copper death-related genes, ultimately constructing a novel ceRNA network and risk model tailored to breast cancer [75]. Li et al. devised a copper death scoring system aimed at forecasting tumor microenvironment infiltration characteristics and gauging the efficacy of immunotherapy [76]. These investigations have primarily concentrated on leveraging copper death mechanisms to forecast the prognosis of breast cancer patients.

In contrast, our study pioneers a novel approach by formulating a breast cancer subtype prediction model grounded in copper death-related ncRNAs. The five key genes utilized in constructing this model exhibited distinct expression patterns across various breast cancer subtypes. This innovative approach offers fresh perspectives for tailoring personalized treatments for breast cancer patients.

This study has some limitations. It only used TCGA dataset for analysis, and the predictive accuracy of the model in real clinical settings needs further verification. We only used an internal dataset for validation. Furthermore, the verification of key genes' roles in breast cancer subtypes necessitates intricate experimental procedures due to the extended duration required for collecting samples of these subtypes. Consequently, this article solely encompasses the present bioinformatics analysis results. Our future endeavors involve substantiating the association between the cuproptosis-related prediction model and breast cancer subtypes through rigorous in vivo or in vitro experiments. Additionally, we aim to delve into the potential mechanisms underlying its impact on breast cancer, thereby enhancing the reliability of our research findings.

In this study, we constructed a breast cancer subtype prediction model using a random forest model and identified five breast cancer subtype biomarkers. We found a strong correlation between the key ncRNA markers in the model and the immune environment, RNA methylation, and angiogenesis. Our findings provide molecular insights into the role of Cuproptosis in predicting breast cancer subtypes and may aid in targeting breast cancer subtypes in clinical practice.

## Supporting information

**S1 Fig. Cuproptosis related genes function and pathway enrichment analysis.**
(TIF)

**S2 Fig. Expression of Cuproptosis-related genes in five subtypes.**
(TIF)

**S1 Table. Breast cancer patient information.**
(XLSX)

**S2 Table. m1A, m5C, m6A genes and angiogenesis-related mRNAs.**
(XLSX)

**S3 Table. Cuproptosis related genes function and pathway enrichment analysis.**
(XLSX)

**S4 Table. 194 ncRNAs differentially expressed in breast cancer.**
(XLSX)

**S5 Table. 94 ncRNAs were significantly associated with differentially expressed ncRNAs.**
(XLSX)

**S6 Table. Expression of key ncRNAs in breast cancer subtypes.**
(XLSX)

**S7 Table. Key ncRNAs in breast cancer subtypes associated with immune infiltration.**
(XLSX)

**S8 Table. Key ncRNAs in breast cancer subtypes associated with RNA methylation modification, and angiogenesis.**
(XLSX)

## Author Contributions

**Conceptualization:** Qurui Wang.

**Data curation:** Qing Xia.

**Formal analysis:** Jinze Shen.

**Funding acquisition:** Shiwei Duan.

**Investigation:** Qurui Wang.

**Writing – original draft:** Qing Xia, Ruixiu Chen.

**Writing – review & editing:** Qing Xia, Xinying Zheng, Qibin Yan, Lihua Du, Hanbing Li, Shiwei Duan.

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
