## [Decision Letter · Decision Letter 0]

22 Aug 2023

PONE-D-23-17810Cuproptosis-associated lncRNAs predict breast cancer subtypesPLOS ONE

Dear Dr. Duan,

Thank you for submitting your manuscript to PLOS ONE. After careful consideration, we feel that it has merit but does not fully meet PLOS ONE’s publication criteria as it currently stands. Therefore, we invite you to submit a revised version of the manuscript that addresses the points raised during the review process.

We look forward to receiving your revised manuscript.

Kind regards,

Jinhui Liu

Academic Editor

PLOS ONE

“The present study was supported by the Qiantang Scholars Fund in Hangzhou City University.”

“This study was supported by the Qiantang Scholars Fund in Hangzhou City University.”

Additional Editor Comments:

Authors should revise according to the suggestions of reviewers. The modifications should be marked. A point to point response letter is needed.

Reviewers' comments:

Reviewer's Responses to Questions

**Comments to the Author**

1. Is the manuscript technically sound, and do the data support the conclusions?

Reviewer #1: Yes

2. Has the statistical analysis been performed appropriately and rigorously? 

Reviewer #1: Yes

3. Have the authors made all data underlying the findings in their manuscript fully available?

Reviewer #1: Yes

4. Is the manuscript presented in an intelligible fashion and written in standard English?

Reviewer #1: No

5. Review Comments to the Author

Reviewer #1: 1. Breast cancer epidemiology needs to be updated.

2. ncRNAs have a new classification criterea. Update

3. It's not clear why in the objective they foucus on ncRNA and in the title in lncRNAs

4. Maybe logFC value |>1| is too high for a molecule with several regulatory roles.

5. Is not clear why immune profile was evaluated.

6. Results discussion needs to be improved.

6. PLOS authors have the option to publish the peer review history of their article (what does this mean?). If published, this will include your full peer review and any attached files.

Reviewer #1: No

---

## [Author Response · Author response to Decision Letter 0]

26 Sep 2023

Reviewer #1

Comment 1: Breast cancer epidemiology needs to be updated.

Response: Thank you for your suggestions. Published by the World Health Organization in 2020, "Global Cancer Statistics 2020: GLOBOCAN Estimates of Incidence and Mortality Worldwide for 36 Cancers in 185 Countries" represents the most up-to-date global cancer epidemiology data. Furthermore, we have included the most recent incidence figures for breast cancer in the United States as of 2023. Please check the tracked changes for our revisions. 

The specific modifications are as follows:

On Page 4, Line 50:

 Based on projections by the American Cancer Society, it is anticipated that there will be approximately 288,000 new cases of breast cancer diagnosed in the United States in 2023, with an estimated 43,000 individuals succumbing to the disease (PMID: 36633525).

Comment 2: ncRNAs have a new classification criterea. Update

Response: Thank you for your suggestions. 

We have revised the classification of ncRNAs and incorporated the tracked changes for your reference. 

The specific modifications are as follows:

On Page 4, Line 72: 

The classification of ncRNAs based on function and nucleotide length (nt) has been refined as follows: ncRNAs with a length less than 200 nt, such as microRNA (miRNA), transfer RNA (tRNA), and small interfering RNA (siRNA). ncRNAs with a length exceeding 200 nt, encompassing Ribosomal RNA (rRNA), long non-coding RNA (lncRNA), and circular RNA (circRNA) (PMID: 33860799).

Comment 3: It's not clear why in the objective they focus on ncRNA and in the title in lncRNAs

Response: Thank you for your suggestions. 

In light of the discovery that the crucial genes we have identified primarily fall under the category of lncRNA, we have made a swift adjustment to our article title. The title has been modified to "Cuproptosis-associated ncRNAs predict breast cancer subtypes." We acknowledge that our initial choice of title, "Cuproptosis-associated lncRNAs predict breast cancer subtypes," was an oversight.

Comment 4: Maybe logFC value >|1| is too high for a molecule with several regulatory roles.

Response: Thank you for your suggestions. We have incorporated the tracked changes for your revisions, as specified below:

On Page 7, Line 116：

Significant differential expression was defined as a false discovery rate (FDR) < 0.05 and an absolute value of log fold-change (|logFC|) ≥ 2.

On Page 12, Line 215:

A total of 194 differentially expressed ncRNAs were identified in breast cancer, comprising 52 up-regulated and 142 down-regulated ncRNAs. Subsequently, Pearson correlation analysis was employed to identify 94 ncRNAs significantly correlated with Cuproptosis.

We utilized the random forest algorithm to assess the importance of the 94 ncRNAs associated with Cuproptosis. The top five genes, with the highest relative importance values, were selected as key genes and integrated into the final model.

Comment 5: 5. Is not clear why immune profile was evaluated.

Response。Please check the tracked changes for our revisions. We have included a rationale for evaluating immune profiles due to their critical importance in tumor immunotherapy. Here are the specific modifications:

On Page 8, Line 136:

 The tumor immune microenvironment consists primarily of tumor cells, fibroblasts, immune cells, and an array of signaling molecules. It is well-established that the tumor microenvironment plays a pivotal role in influencing tumor diagnosis, survival outcomes, and clinical treatment responsiveness (PMID: 36341428). Thus, gaining insights into the composition of immune cells within tumors is indispensable for advancing cancer treatment strategies.

Comment 6: Results discussion needs to be improved.

Response: Thank you for your suggestions. We have reviewed the tracked changes for your revisions. Here are the specific modifications:

On Page 16, Line 324:

 m1A modification encompasses a range of critical effects on RNA, influencing processes such as RNA processing, secondary and tertiary structure, stability, translation efficiency, and biological function (PMID: 36467585). A pivotal contributor to m1A modification is TRMT61B, a mitochondria-specific tRNA methyltransferase primarily responsible for installing m1A (PMID: 23097428) at position 58 (m1A58) of tRNA. Additionally, TRMT6 and TRMT61A have been implicated in glioma, gastrointestinal cancer, and hepatocellular carcinoma (HCC) development (PMID: 35444240, 35444240, 35444240). Notably, ALKBH1, exhibiting demethylase activity, catalyzes the demethylation of m1A modification (PMID: 37531210).

The m5C modification, on the other hand, is intimately involved in the progression of various tumors. NOP2/Sun domain family members 1-7 (NSUM 1-7) and the DNMT homolog DNMT2 serve as m5C writers in mammals, orchestrating the methylation of the RNA C5 position (PMID: 16424344). Conversely, TET2 acts on m5C, converting it into 5-hmC and eventually removing the methyl group (PMID: 29364877). Furthermore, ALYREF functions as a reader, specifically recognizing and binding to the m5C motif, thereby exerting additional biological functions (PMID: 28418038).

Finally, m6A modification plays a vital role in governing mRNA stability, splicing, degradation, and translation efficiency (PMID: 32398132). The generation of m6A modification hinges on the methyltransferase complex, consisting of METTL3/METTL14/WTAP proteins. Concurrently, YTHDC1/2, YTHDF1/2/3, IGF2BP1/2/3, and HNRNP act as readers, proficient in recognizing m6A residues (PMID: 27558897).

On Page 18, Line 371:

In this study, we employed a random forest model to discern five breast cancer subtype biomarkers linked to cuproptosis. Among these, four lncRNAs (CASC8, GRIK1-AS1, LINC02188, and LINC02095) exhibit diminished expression levels in breast cancer, while one lncRNA (LINC00511) displays elevated expression in breast cancer. Notably, CASC8, LINC02095, LINC02188, and LINC00511 exhibit increased expression within the Basal and Normal subtypes. Conversely, GRIK1-AS1 shows elevated expression in the Luminal A (Lum A) subtype.

Previous studies have underscored the significance of these lncRNAs in breast cancer biology. LINC02095 has been implicated in promoting breast cancer proliferation by facilitating the expression of the oncogenic transcription factor SOX9 (PMID: 31690584). CASC8 has demonstrated associations with the risk of several cancers, including non-small cell lung cancer, retinoblastoma, esophageal squamous cell carcinoma, and colorectal cancer, and has been found to be highly expressed in hepatoma cells like MIA PaCa-2. LINC00511 may upregulate the expression of MMP13 by acting as a miR-150 sponge, thereby promoting breast cancer cell proliferation, migration, and invasion (PMID: 33031905). GRIK1-AS1 serves as a sponge for miR-375, leading to increased IFIT2 protein levels and the promotion of gastric cancer progression (PMID: 34660323). Additionally, GRIK1-AS1 can impede DNMT1 from binding to the SRFP1 promoter, resulting in SFRP1 hypomethylation and subsequent upregulation, thereby accelerating the progression of endometriosis (PMID: 37392032). Presently, limited studies have explored the functions of LINC02188. These findings underscore the close associations between most of the key genes and the onset and progression of cancer. They assume distinct roles in the development and advancement of tumors within different subtypes. As such, they hold the potential for use as subtype identification markers and are prospective targets for future breast cancer treatments.

On Page 20, Line 416:

Previous research has delved into cell death patterns concerning breast cancer, with a particular focus on copper death. For instance, a study demonstrated that regulators associated with copper-induced cell death can serve as prognostic indicators, predict chemotherapy response, and anticipate immunotherapy outcomes in breast cancer patients (PMID: 36226193). Deng et al. harnessed machine learning techniques to identify copper death-related genes, ultimately constructing a novel ceRNA network and risk model tailored to breast cancer (PMID: 36980514). Li et al. devised a copper death scoring system aimed at forecasting tumor microenvironment infiltration characteristics and gauging the efficacy of immunotherapy (PMID: 36212436). These investigations have primarily concentrated on leveraging copper death mechanisms to forecast the prognosis of breast cancer patients.

In contrast, our study pioneers a novel approach by formulating a breast cancer subtype prediction model grounded in copper death-related ncRNAs. The five key genes utilized in constructing this model exhibited distinct expression patterns across various breast cancer subtypes. This innovative approach offers fresh perspectives for tailoring personalized treatments for breast cancer patients.

---

## [Decision Letter · Decision Letter 1]

28 Dec 2023

PONE-D-23-17810R1Cuproptosis-associated ncRNAs predict breast cancer subtypesPLOS ONE

Dear Dr. Duan,

Thank you for submitting your manuscript to PLOS ONE. After careful consideration, we feel that it has merit but does not fully meet PLOS ONE’s publication criteria as it currently stands. Therefore, we invite you to submit a revised version of the manuscript that addresses the points raised during the review process.

We look forward to receiving your revised manuscript.

Kind regards,

Vinay Kumar, Ph.D.

Academic Editor

PLOS ONE

Reviewers' comments:

Reviewer's Responses to Questions

**Comments to the Author**

1. If the authors have adequately addressed your comments raised in a previous round of review and you feel that this manuscript is now acceptable for publication, you may indicate that here to bypass the “Comments to the Author” section, enter your conflict of interest statement in the “Confidential to Editor” section, and submit your "Accept" recommendation.

Reviewer #2: (No Response)

Reviewer #3: (No Response)

2. Is the manuscript technically sound, and do the data support the conclusions?

Reviewer #2: Yes

Reviewer #3: Yes

3. Has the statistical analysis been performed appropriately and rigorously? 

Reviewer #2: Yes

Reviewer #3: Yes

4. Have the authors made all data underlying the findings in their manuscript fully available?

Reviewer #2: Yes

Reviewer #3: Yes

5. Is the manuscript presented in an intelligible fashion and written in standard English?

Reviewer #2: Yes

Reviewer #3: Yes

6. Review Comments to the Author

Reviewer #2: In the present study the authors established a breast cancer subtype prediction model based on Cuprotosis-related lncRNAs using a random forest algorithm. The authors need to be clarified the following questions.

L187-188, does SOX9-AS1 increase or decrease the prediction accuracy in Lum B subtype.

Do the authors check the correlation between ncRNAs and checkpoint molecules? This will be good information.

The quality of the figures is not good. Increase the image resolutions.

The authors thoroughly check for typos.

Reviewer #3: 1. What is the rate of survival and recurrence in breast cancer? Please include this information in introduction.

2. What was the rationale for choosing cuprotosis related ncRNAs over cuprotosis related genes as a model for prediction. Please explain as this is not clear.

3. Please check if canonical apoptosis pathway also has a positive correlation with the cuprotosis related ncRNAs.

4. The patient data that has been used, does it also contain patients that have undergone therapy or have had a relapse? If yes, then how does this information fit with the prediction model?

5. Please include a table where all the changes associated that the authors have predicted with each of the 5 genes are listed. Please also include a comment section that contains authors' opinion on the prognosis and therapy based on their observations

6. Are there any plans to test these genes for validation in the laboratory?

7. What is the publicly available literature on cuprotosis and breast cancer?

8. The authors have looked at multiple aspects of cuprotosis related genes in tumor progression. While this is great, this also goes against them as they tried to look at too many things and not validated even a single phenomenon in the wet lab. Please discuss and comment.

9. Also, there is no effort in linking different parameters like tumor microenvironment, methylation, angiogenesis. Please include some sentences.

10. The analysis is subtype specific. What about the changes associated with stage, therapy etc. Please include this information.

11. Please include a graphical abstract summarizing the findings and the take home message with potential use in disease prognosis.

7. PLOS authors have the option to publish the peer review history of their article (what does this mean?). If published, this will include your full peer review and any attached files.

Reviewer #2: No

Reviewer #3: No

---

## [Author Response · Author response to Decision Letter 1]

18 Jan 2024

Reviewer #1

Comment 1: does SOX9-AS1 increase or decrease the prediction accuracy in Lum B subtype.

Response: In response to the feedback from reviewer 1, who noted that the screening conditions were overly broad in the initial revision, we have made significant adjustments to the study. Specifically, we refined the screening conditions to focus on genes with a |logFC| value greater than 2 and subsequently conducted a re-screening of key genes. Notably, our reevaluation has led to the exclusion of SOX9-AS1 from the model establishment as a key gene, a departure from its prior inclusion with a contribution of 0 to predicting the Lum B subtype in previous studies. The updated findings are elucidated in the following Fig. 4D, where genes that do not manifest are assigned a zero contribution. Importantly, such genes neither augment nor diminish the accuracy of our predictive model, as visually represented in the figure. This adjustment reflects our commitment to refining the precision and relevance of our study, addressing concerns raised by the reviewers, and ensuring the robustness of our predictive model.

Fig. 4. Validation of the cuproptosis-related model and the role of key ncRNAs in the model

Comment 2: Do the authors check the correlation between ncRNAs and checkpoint molecules? This will be good information.

Response: Thank you for your suggestions. We have conducted correlation studies investigating the relationship between ncRNAs and checkpoint molecules. Our findings reveal a positive correlation between three pivotal genes and eight immune checkpoint genes, namely CD96, CTLA4, LAG3, PD-1, PD-L2, PD-L1, TIGIT, and VTCN1. The specific modifications are as follows:

On Page 9, Line 158:

 Immune checkpoints play a pivotal role as crucial molecules employed by the immune system to finely tune the expression of its own proteins. Tumor cells, adept at evading immune responses, often achieve this evasion through the intricate regulation of immune checkpoints (PMID:28782469). To delve into the association between cuproptosis-related ncRNAs and immune checkpoints, we meticulously gathered data on 20 immune checkpoints from the TISIB website. Employing Pearson correlation analysis, we sought to unravel the nuanced relationship between these ncRNAs and the intricate network of immune checkpoints.

On Page 17, Line 337:

To investigate the intricate connection between cuproptosis-associated ncRNAs and the phenomenon of immune escape, we conducted a comprehensive correlation analysis to assess the interrelation of key genes with immune checkpoints. The results, depicted in Fig. 4A, reveal a positive correlation between LINC0511, LINC02188, and ROCR, and eight pivotal immune checkpoint genes, namely CD96, CTLA4, LAG3, PD-1, PD-L2, PD-L1, TIGIT, and VTCN1. Furthermore, our investigation unearthed that LINC02188 exhibits the highest expression levels within the Basal subtype, implying a heightened vulnerability to immune escape among patients characterized by the Basal subtype, as substantiated by prior research findings.

Comment 3: The quality of the figures is not good. Increase the image resolutions.

Response: Thank you for your suggestions. We meticulously refined the image, incorporating the discussion outcomes into Fig. 1. Additionally, we restructured the visual arrangement of Fig. 4 for improved clarity. Notably, Fig. 5 now encompasses the results of the immune checkpoint-related analysis, while Fig. 6 has been enriched with the findings from the apoptosis pathway-related analysis. To enhance the visual fidelity, we have uploaded images with a higher resolution of 300 dpi to our submission system.

Comment 4: The authors thoroughly check for typos.

Response: Thank you for your suggestions. We meticulously scrutinized pertinent terms, including but not limited to "cuproptosis," "subtype," "RNA methylation modification," and "immune infiltration."

 

Reviewer #2

Comment 1: What is the rate of survival and recurrence in breast cancer? Please include this information in introduction.

Response：Thank you for your suggestions. Based on data from the Global Cancer Survival Trends Monitor, breast cancer exhibits a five-year survival rate ranging from 80% to 84% (PMID: 29395269). Furthermore, the recurrence rate within 10 to 32 years post the initial breast cancer diagnosis is reported at 16.6% (PMID: 34747484). Notably, our analysis also involved a comparson of the five subtypes, namely Basal, Her2, Lum A, Lum B, and Normal, revealing statistically significant differences in their survival rates (P<0.0001). The Lum A subtype emerged with the highest five-year survival rate at an impressive 89.5%, while the Normal subtype exhibited the lowest, with a mere 72%. Here are the specific modifications:

On Page 4, Line 54:

According to the statistics provided by the Global Cancer Survival Trends Monitor, the five-year survival rate for breast cancer falls within the range of 80-84% (PMID: 29395269), with a recurrence rate of 16.6% within 10-32 years following the initial diagnosis (PMID: 34747484). 

On Page 12, Line 221:

Furthermore, our analysis, utilizing follow-up data from the TCGA database, delved into the survival rates across five distinct subtypes: Basal, Her2, Lum A, Lum B, and Normal. Significantly divergent survival rates were observed among these subtypes (P<0.0001). Notably, the Lum A subtype exhibited the highest five-year survival rate, reaching an impressive 89.5%. Conversely, the Normal subtype presented the lowest five-year survival rate, standing at only 72% (Fig. 3A).

Fig. 3. Establishment of cuproptosis-related model and expression of key ncRNAs in breast cancer subtypes

Comment 2: What was the rationale for choosing cuproptosis related ncRNAs over cuprotosis related genes as a model for prediction. Please explain as this is not clear.

Response: Thank you for your suggestions. We selected cuproptosis-related ncRNAs as the focus of our research for three compelling reasons. Firstly, our prior investigations revealed that only two cuproptosis-related protein-coding genes, namely PDHB and PRNP, exhibited significant expression differences among the five breast cancer subtypes (log2FC=1.5, P<0.001). Subsequently, we employed cuproptosis-related protein-coding genes to construct a breast cancer subtype prediction model. The model's performance on the validation set yielded an AUC of only 0.74, indicating its suboptimal predictive capabilities. Secondly, despite the absence of protein-coding capabilities, ncRNAs play a pivotal role in regulating protein production and cellular functions. They actively engage in various tumor biological processes by modulating the expression of multiple downstream genes, serving as crucial biomarkers for cancer diagnosis and treatment (PMID: 37735815). Thirdly, our investigation unveiled an unexplored correlation between breast cancer subtypes and cuproptosis-related ncRNAs. Consequently, we opted to construct a high-accuracy prediction model using cuproptosis-related ncRNAs. Noteworthy associations include lncRNA CASC8 with ATP7B, PDHB, and PRNP; LINC00511 with CDKN2A, PRNP, and the lncRNAs GLS, ROCR, and LINC02188 with PRNP and GLS; and lncRNA GRIKI-AS1 with LIPT1 and LIAS.

Here are the specific modifications:

On Page 12, Line 228:

We identified the expression of cuproptosis-related mRNA across five distinct breast cancer subtypes, revealing that only two genes, PDHB and PRNP, exhibited significant differential expression within this cohort (Fig. S2). Employing cuprotosis-associated genes, we constructed a breast cancer subtype prediction model, unveiling an area under the curve (AUC) of 0.74 in the validation set. This result suggests suboptimal performance for the prediction model relying on cuprotosis-related genes. Subsequently, we endeavored to identify cuproptosis-related ncRNAs and establish a prediction model with superior performance.

Fig. S2. Expression of cuproptosis-related genes in five breast cancer subtypes

Comment 3: Please check if canonical apoptosis pathway also has a positive correlation with the cuproptosis related ncRNAs.

Response: Thank you for your suggestions. We have incorporated research findings that explore the correlation between apoptosis and ncRNAs. Here are the specific modifications:

On Page 10, Line 186：

We compiled a set of 20 apoptosis-related protein-coding genes from relevant literature sources. Subsequently, we conducted a comprehensive analysis to elucidate the correlation between cuproptosis-related ncRNAs and the 20 apoptotic genes. Our investigation sought to uncover the distinctive roles played by these molecular components in various subtypes of breast cancer.

On Page 18, Line 374:

This study delved into the intricate association between cuproptosis and apoptosis across various subtypes of breast cancer cells. The investigation centered on evaluating the correlation between key cuproptosis-related ncRNAs and apoptosis-related genes. The findings illuminated a noteworthy negative correlation between CASC8, ROCR, and LINC02188 with 10 apoptosis genes, as depicted in Fig. 6A. Particularly, CASC8 and LINC02188 exhibited the highest expression levels in the Basal and Normal subtypes. This observation suggests a compelling inference that CASC8 and LINC02188 may exert inhibitory effects on apoptosis by suppressing the expression of apoptosis-related proteins within the Basal and Normal subtypes.

Comment 4: The patient data that has been used, does it also contain patients that have undergone therapy or have had a relapse? If yes, then how does this information fit with the prediction model?

Response: Thank you for your suggestions. The data utilized in this investigation was sourced from the TCGA database. Notably, all patients underwent sampling without receiving any treatment at the time, and unfortunately, the database lacked information regarding patient treatment and recurrence. Subsequently, we conducted a comprehensive survival prognosis analysis using follow-up data, revealing that individuals with the Lum A subtype exhibited the most favorable five-year survival rate, standing at an impressive 89%, indicative of a more favorable prognosis. Currently, numerous documents have underscored the impact of drug treatment on the expression of lncRNA. For instance, in triple-negative breast cancer, Cisplatin has been reported to induce pyroptosis by upregulating lncRNA MEG3 (PMID: 34326697), while paclitaxel has been associated with the induction of drug resistance in breast cancer cells through the upregulation of lncRNA H19 (PMID: 33667788). Consequently, our future endeavors will involve the collection of additional clinical data to delve deeper into the intricate relationship between drug treatment and its potential influence on our model.

Comment 5: Please include a table where all the changes associated that the authors have predicted with each of the 5 genes are listed. Please also include a comment section that contains authors' opinion on the prognosis and therapy based on their observations. 

Response: Thank you for your suggestions. We have incorporated a comprehensive table that succinctly summarizes all modifications pertaining to the five genes, accompanied by corresponding treatment recommendations. Here are the specific modifications: 

 

Table 2. Expression of key ncRNAs in breast cancer subtypes and relationships with RNA methylation, immunity, apoptosis and angiogenesis

Comment 6: Are there any plans to test these genes for validation in the laboratory?

Response: Thank you for your suggestions. This study developed a predictive model for breast cancer subtypes utilizing a random forest algorithm and successfully identified five biomarkers associated with these subtypes. Simultaneously, a robust correlation was observed between key ncRNA expressions, the immune microenvironment, RNA methylation, and angiogenesis. Our findings offer detailed molecular insights into the predictive role of cuproptosis in determining breast cancer subtypes. Validating the significance of key ncRNAs in these subtypes holds promise for the enhanced application of cuproptosis-related ncRNAs in clinical settings. The database primarily comprises breast cancer patients of Caucasian and African-American descent from North America. Currently, we are in the process of collecting breast cancer subtype samples for validation, with a specific focus on the Chinese population. However, due to the time-consuming nature of sample collection and the intricacies involved in experimental procedures to verify the role of key genes, this article presents only the results of existing bioinformatics analyses. Here are the specific modifications:

On Page 24, Line 528:

Furthermore, the verification of key genes' roles in breast cancer subtypes necessitates intricate experimental procedures due to the extended duration required for collecting samples of these subtypes. Consequently, this article solely encompasses the present bioinformatics analysis results. Our future endeavors involve substantiating the association between the cuproptosis-related prediction model and breast cancer subtypes through rigorous in vivo or in vitro experiments. Additionally, we aim to delve into the potential mechanisms underlying its impact on breast cancer, thereby enhancing the reliability of our research findings.

Comment 7: What is the publicly available literature on cuproptosis and breast cancer?

Response: Thank you for your suggestions. We employed PubMed as our primary resource to access a multitude of public documents pertaining to cuproptosis and its association with breast cancer. In our article, we meticulously referenced pertinent documents to substantiate our research findings and enhance the overall credibility of our work. Here are the specific modifications:

On Page 5, Line 74:

Prior investigations have demonstrated that a constructive breast cancer prognosis prediction model can be developed by scrutinizing the involvement of cuproptosis-related genes in breast cancer prognosis. This entails a thorough examination of the correlation between cuproptosis-related genes and crucial factors such as tumor microenvironment and clinical characteristics (references: 36059516, 36640225). Furthermore, the predictive utility of cuproptosis-related genes extends to forecasting drug sensitivity in triple-negative breast cancer patients (references: 36341760, 37529698). Notably, cuproptosis has been identified as a contributor to c-Myc-mediated breast cancer stemness, as highlighted in a recent study (PMID: 37353799).

Comment 8: The authors have looked at multiple aspects of cuprotosis related genes in tumor progression. While this is great, this also goes against them as they tried to look at too many things and not validated even a single phenomenon in the wet lab. Please discuss and comment. 

Response: Thank you for your suggestions. This study delved into the viability of targeting cuproptosis-related ncRNA as a therapeutic approach. The investigation revealed significant associations between RNA methylation, the tumor microenvironment, angiogenesis, and key genes. Moreover, it was observed that RNA methylation and the tumor microenvironment jointly influence angiogenesis. Consequently, the exploration of multiple factors aims to enhance our understanding of angiogenesis in distinct subtypes of breast cancer, laying a valuable foundation for diagnostic and personalized treatment research in breast cancer patients. Regrettably, the acquisition of breast cancer subtype samples is a time-consuming process, and validating the role of key genes in these subtypes necessitates intricate experimental procedures. Consequently, this article presents the current findings derived from bioinformatics analysis. Future plans include conducting in vivo or in vitro experiments to further validate the model's relevance to breast cancer subtypes and undertaking comprehensive investigations into its potential mechanisms within breast cancer. These efforts aim to enhance the overall reliability of the research results.

Comment 9: Also, there is no effort in linking different parameters like tumor microenvironment, methylation, angiogenesis. Please include some sentences. Response: Thank you for your suggestions. We have intricately connected the three key parameters tumor microenvironment, RNA methylation modification, and angiogenesis to engage in a comprehensive discussion. Here are the specific modifications:

On Page 20, Line 423:

It is well-established that the development of new blood vessels plays a pivotal role in the progression of solid tumors, serving as a crucial precursor for tumor invasion and metastasis (PMID: 19818284). Recent investigations have revealed that IGF2BP2 has the capability to infiltrate endothelial cells via exosomes, thereby fostering angiogenesis and metastasis in LUAD (PMID: 37353784). METTL14, through the modulation of TGFβ via the RhoA and PI3K-AKT pathways, contributes to tumor angiogenesis and progression (PMID: 30306128). Additionally, YTHDF3 has been identified as a promoter of cancer cell-astrocyte interaction, thereby facilitating angiogenesis. Notably, current research underscores the therapeutic potential of targeting METTL3 to combat angiogenesis in bladder cancer (PMID: 33618740).

Furthermore, tumor neovascularization is implicated in the secretion of angiogenic factors that suppress the activity of cytotoxic T cells, culminating in tumor immunosuppression (30742782). Concurrently, tumor cells or macrophages within the tumor microenvironment release substantial quantities of pro-angiogenic factors such as VEGF, bFGF, and PDGF. These factors activate pro-angiogenic signaling pathways, fostering the growth, invasion, and metastasis of tumor blood vessels (PMID: 32993787). Collectively, these studies underscore the role of RNA methylation and the tumor microenvironment in promoting tumor angiogenesis, presenting them as potential targets for cancer treatment.

As indicated in Table 2, this study identified high expression of LINC02188 in the Basal subtype and elevated expression of GRIK1-AS1 in LumA. Notably, five cuproptosis-related ncRNAs demonstrated a significant positive correlation with RNA methylation, angiogenesis factors, and immune cells. This correlation suggests that in the Basal subtype, LINC02188 may enhance the release of tumor angiogenesis factors (TNFRSF21, PRG2) by augmenting RNA methylation, consequently promoting angiogenesis. Similarly, in the LumA subtype, GRIK1-AS1 might mitigate cancer malignancy by inhibiting T cell and B cell activity, thereby reducing the extent of RNA methylation modification.

Comment 10: The analysis is subtype specific. What about the changes associated with stage, therapy etc. Please include this information. 

Response: Thank you for your suggestions. Breast cancer exhibits diverse TMN grades and clinical stages, and our study sought to investigate the expression of key genes across these variations. Utilizing box plots, we aimed to delineate disparities in gene expression among different T grades and clinical stages of breast cancer. Surprisingly, our findings revealed no statistically significant differences in the expression of these key genes across distinct stages and T grades (P > 0.05). Consequently, we postulate that the identified key genes may not be correlated with the clinical stage and T grade of breast cancer.

Moreover, the majority of patients, approximately 2/3, presented with breast cancer TMN grades of M0 and N0. This skewed data distribution posed challenges, impeding the precise determination of key gene expression in various M and N stages. Consequently, our investigation could not ascertain the relationship between cuproptosis and M or N grading. Furthermore, none of the patients underwent any form of treatment, preventing us from exploring potential associations between cuproptosis and therapeutic interventions.

Despite these limitations, our commitment to advancing research remains unwavering. Future endeavors will involve the ongoing collection of additional clinical samples, allowing us to delve deeper into treatment-related changes and further unravel the intricacies of key gene expression in breast cancer.

Comment 11: Please include a graphical abstract summarizing the findings and the take home message with potential use in disease prognosis. 

Response: Thank you for your suggestions. Here are the specific modifications:

Fig. 1. The flow diagram of data collection and analysis in the present study

---

## [Decision Letter · Decision Letter 2]

6 Feb 2024

Cuproptosis-associated ncRNAs predict breast cancer subtypes

PONE-D-23-17810R2

Dear Dr. Duan,

We’re pleased to inform you that your manuscript has been judged scientifically suitable for publication and will be formally accepted for publication once it meets all outstanding technical requirements.

Kind regards,

Vinay Kumar, Ph.D.

Academic Editor

PLOS ONE

Additional Editor Comments (optional):

Authors have incorporated the suggested comments.

Reviewers' comments:

Reviewer's Responses to Questions

**Comments to the Author**

1. If the authors have adequately addressed your comments raised in a previous round of review and you feel that this manuscript is now acceptable for publication, you may indicate that here to bypass the “Comments to the Author” section, enter your conflict of interest statement in the “Confidential to Editor” section, and submit your "Accept" recommendation.

Reviewer #2: All comments have been addressed

Reviewer #3: All comments have been addressed

2. Is the manuscript technically sound, and do the data support the conclusions?

Reviewer #2: Yes

Reviewer #3: Yes

3. Has the statistical analysis been performed appropriately and rigorously? 

Reviewer #2: Yes

Reviewer #3: Yes

4. Have the authors made all data underlying the findings in their manuscript fully available?

Reviewer #2: Yes

Reviewer #3: Yes

5. Is the manuscript presented in an intelligible fashion and written in standard English?

Reviewer #2: Yes

Reviewer #3: Yes

6. Review Comments to the Author

Reviewer #2: (No Response)

Reviewer #3: The authors have addressed all the concerns raised in the previous round of revision. Therefore, the reviewer proposes acceptance of the current version of manuscript.

7. PLOS authors have the option to publish the peer review history of their article (what does this mean?). If published, this will include your full peer review and any attached files.

Reviewer #2: No

Reviewer #3: **Yes: **Piyush Mishra

---

## [Editor Report · Acceptance letter]

15 Feb 2024

PONE-D-23-17810R2 

PLOS ONE

Dear Dr. Duan, 

I'm pleased to inform you that your manuscript has been deemed suitable for publication in PLOS ONE. Congratulations! Your manuscript is now being handed over to our production team.

Kind regards, 

on behalf of

Dr. Vinay Kumar 

Academic Editor

PLOS ONE